# Honey Bee Colony Population Daily Loss Rate Forecasting and an Early Warning Method Using Temporal Convolutional Networks

**DOI:** 10.3390/s21113900

**Published:** 2021-06-04

**Authors:** Thi-Nha Ngo, Dan Jeric Arcega Rustia, En-Cheng Yang, Ta-Te Lin

**Affiliations:** 1Department of Biomechatronics Engineering, National Taiwan University, Taipei 10617, Taiwan; d03631008@ntu.edu.tw (T.-N.N.); d05631006@ntu.edu.tw (D.J.A.R.); 2Department of Entomology, National Taiwan University, Taipei 10617, Taiwan; ecyang@ntu.edu.tw; 3Graduate Institute of Brain and Mind Sciences, National Taiwan University, Taipei 10617, Taiwan

**Keywords:** time series forecasting, monitoring system, early warning, temporal convolution networks, population daily loss rate

## Abstract

The population loss rate of a honey bee colony is a critical index to verify its health condition. Forecasting models for the population loss rate of a honey bee colony can be an essential tool in honey bee health management and pave a way to early warning methods in the understanding of potential abnormalities affecting a honey bee colony. This work presents a forecasting and early warning algorithm for the population daily loss rate of honey bee colonies and determining warning levels based on the predictions. Honey bee colony population daily loss rate data were obtained through embedded image systems to automatically monitor in real-time the in-and-out activity of honey bees at hive entrances. A forecasting model was trained based on temporal convolutional neural networks (TCN) to predict the following day’s population loss rate. The forecasting model was optimized by conducting feature importance analysis, feature selection, and hyperparameter optimization. A warning level determination method using an isolation forest algorithm was applied to classify the population daily loss rate as normal or abnormal. The integrated algorithm was tested on two population loss rate datasets collected from multiple honey bee colonies in a honey bee farm. The test results show that the forecasting model can achieve a weighted mean average percentage error (*WMAPE*) of 17.1 ± 1.6%, while the warning level determination method reached 90.0 ± 8.5% accuracy. The forecasting model developed through this study can be used to facilitate efficient management of honey bee colonies and prevent colony collapse.

## 1. Introduction

Honey bees (*Apis mellifera*) are essential for global food production [1]. In the last decades, some regions of the world have suffered from a significant loss in the number of active honey bee colonies [2]. The decline of honey bee colonies is partly attributed to a phenomenon known as Colony Collapse Disorder (CCD) [1,2]. To maintain healthy and thriving honey bee colonies, it is essential for beekeepers to regularly monitor the status of the bee hives. Currently, most beekeepers manually estimate the health of a honey bee colony in a relatively time-consuming and imprecise approach by beehive inspection and honey bee flight activity observation at the hive entrance. Automated monitoring system are highly desirable to have a rapid and quantitative access to the health status information of honey bee hives.

Various systems for automatic in-hive and hive-entrance monitoring of honey bee activities have been developed. Reported systems use infrared sensors [3], radio frequency identification (RFID) tags and readers [4,5], and imaging sensors [6]. In our recent work, we described an embedded image monitoring system for automatic and highly reliable counting of incoming and outgoing honey bees at the hive entrance [7]. The obtained access to in-time and quantitative data opens a path to the development of forecasting models as an essential tool for beehive management, which is discussed here. When the daily population loss rate exceeds a critical threshold, the system can release an automated alert of a required and immediate intervention by the beekeeper to promptly prevent honey bee colony collapse.

An automatic system for monitoring and forecasting honey bee activity and daily loss rate has several advantages. An automated system reduces labor costs in maintaining the health of a colony [1] and enables the identification of abandoned or collapsed hives [8]. It can also detect quantitative deviations from a predicted normal status, as through the invasion of predators such as a hornet, the presence of pesticides in harvested areas, or strong fluctuations in environmental conditions. Forecasting models may also be used to investigate the environmental variables that largely affect the health of a honey bee colony [9].

The collection of long-term activity data on honey bee colonies is difficult. Only few studies related to forecasting honey bee activity are reported. An early detection system was developed by Ferrari et al. [10] for swarm monitoring by recording the auditory activity of honeybees. A microphone and a temperature and humidity sensor were installed inside the beehive. The recorded audio signals were used to predict swarming. In the work of Clarke and Robert [9], a model was developed to predict the hourly egress rate of honey bees from local weather data such as: temperature, solar radiation, atmospheric pressure, humidity, rainfall, wind direction and wind speed. About 78% of the observed variation in honey bee activity could be explained by variations in temperature and solar radiation. In a study by Bagheri and Mirzaie [11], a mathematical model was developed to predict the effect of pollen on honey bee colony failure, whereas Gomes et al. [12] developed a method for forecasting the hourly activity level of honey bees using a recurrent neural network (RNN). Environmental factors, such as temperature, solar radiation and barometric pressure, together with the history of honey bee activity, were used to train the forecasting model producing a root mean square error of 0.2.

There are several possible ways in performing time series forecasting. Statistical methods include autoregressive integrated moving average (ARIMA) [13] and Bayesian algorithms [14]. Machine learning techniques, including random forest and support vector machine, are promising explicitly for more complex datasets [15]. Honey bee daily loss rate is a complex and dynamic model involving different variables inside and outside the beehive. Statistical and machine learning approaches may not be adequate for complex pattern recognition. Increasingly, more researchers were able to use deep learning models for obtaining accurate and reliable forecasting results. Most deep learning models for time-series forecasting comprise recurrent neural networks (RNN). RNNs are based on mechanisms such as gated recurrent units (GRU) [16] and long-short term memory (LSTM) [17,18]. Another form of neural network for time-series forecasting is the temporal convolutional neural network (TCN) [19]. TCN was first introduced by Lea et al. [20] for video-based action segmentation. TCN provides a unified approach for hierarchically capturing data relationships with different levels of information.

Warning level categories to classify forecasted loss rates can provide insightful information to beekeepers. In the work of Bayuadji et al. [21], a flood warning level was determined by fitting flood and rainfall data to a logistic regression model. Thereby, pre-set probability thresholds classified rainfall into three warning levels. On the other hand, Zhang et al. [22] used K-means clustering algorithm to classify warning levels for recorded vegetable insect pest populations from level I to level IV, being the highest warning level. Up to now, there is no related work for defining warning levels for honey bee collapse.

This work aims to develop a honey bee colony daily population loss rate forecasting method, and based on the results, define warning levels. To meet these goals, specific objectives to be fulfilled included: (1) developing a TCN model for forecasting honey bee colony daily population loss rate; (2) optimizing the developed TCN model by finding its ideal hyperparameters and determining its best input features; (3) implementing an outlier detection method, such as an isolation forest algorithm, to classify the forecasted daily population loss rate as normal or abnormal; and (4) evaluating the developed TCN model and isolation forest algorithm based on test datasets. The developed method can be used by beekeepers, entomologists, and agronomists to obtain fast, essential information about the status and abnormality in honey bee colonies.

## 2. Materials and Methods

### 2.1. Data Collection

An automated system was used for recording the traffic at the beehive entrance and monitoring local environmental conditions. Each set of the system included an NVIDIA Jetson TX2 (NVIDIA Corporation, Santa Clara, CA, USA) for image monitoring in the beehive entrance, and a Raspberry Pi 3 (Raspberry Pi Foundation, Cambridge, UK) embedded system for acquiring sensor data including temperature, humidity, light intensity, and rain level. A black acrylic observation box, which included a web camera and red LED lighting board, was attached to the beehive entrance. Technical details are discussed in our previous work [7].

Data collected by the image monitoring system included daily incoming and outgoing counts, daily differences in incoming and outgoing counts, daily population loss rate, temperature, and relative humidity. The daily population loss rate (*LR*) of a honey bee colony was calculated using Equation (1):(1)$$LR\left(t\right)=\frac{C_{out}\left(t\right)-C_{in}\left(t\right)}{C_{out}\left(t\right)}$$
where *C_in_* (*t*) and *C_out_* (*t*), as functions of time (*t*), are the daily incoming and outgoing counts, respectively. This equation was used in our previous work to assess the effects of pesticides on honey bee colonies [7].

Local weather data were wind speed, wind speed gust, wind direction, wind direction gust, precipitation, and the duration of precipitation and collected from the open-source data platform of the Taiwan Central Weather Bureau. The local weather data were cross verified with the rain level, temperature, and humidity sensor data collected by the monitoring system; as they showed similar values, they were used as additional input features of the forecasting model. All collected data were used for feature selection of the forecasting model and analysis. The nomenclature of the data is shown in Table 1.

### 2.2. Experimental Setup

Experimental data were collected from two experiments in a honey bee farm located in Hsinchu City, Taiwan. The details of each experiment are summarized in Table 2. The honey bees studied in this research were *Apis mellifera*. In each experiment, four healthy colonies with four honey bee combs and one queen each were prepared. The four replicated beehives were used to test the adaptability of the proposed forecasting algorithm. The hives were checked every two weeks to ensure the healthy condition of the colony. The data obtained from the two experiments were referred to as dataset E_A*n*_ and E_B*n*_, respectively, where *n* is the beehive number.

### 2.3. Honey Bee Colony Population Loss Rate Forecasting and Early Warning Algorithm

The flowchart of the forecasting and early warning algorithm is presented in Figure 1. The algorithm was implemented using Python 3.5 with the support of Keras v2.2.6 deep learning library [23] and SciKit-Learn machine learning library [24].

#### 2.3.1. Data Pre-Processing

*LR* data collected by the image monitoring system were pre-processed by interpolation and data normalization before feeding into the TCN forecasting model. Sample results are shown in Figure 1. Interpolation is a mathematical method that fits a function to a dataset and uses the function to fill in missing data based on the nearest past and future values. Each set of missing data was filled in with the mean computed from its four nearest values. Afterwards, the completed data were normalized by scaling the input feature into values from 0 to 1 based on the minimum and maximum values of each input feature, respectively. Dataset E_A_ had five days of missing data while dataset E_B_ had eight days of missing data, both due to power outage and maintenance of the image monitoring system.

#### 2.3.2. Temporal Convolutional Network for Daily Population Loss Rate Forecasting

A temporal convolutional network (TCN) [20] was used to forecast the future honey bee daily population loss rate and predict potential colony collapse. The key characteristic of TCNs is its usage of convolutions. Convolutions are causal and does not depend on any future timestep data. Unlike other deep learning models for forecasting like GRU and LSTM. TCN has longer memory and can process time series data of any length and generate similar long predictions [25]. For these reasons, TCN was selected as the forecasting model in this work.

The TCN forecasting model comprised a series of blocks, which individually contained a sequence of convolutional layers. Each layer was composed of dilated convolutions, associated with a dilation factor *d*, with rectified linear unit (ReLU) as non-linear activation function. Dilation introduces a fixed step between every adjacent filter. Larger dilations and larger filters of size *k* effectively expand the receptive field [19,20]. In these convolutions, exponential increments in the value of *d* increases the depth of the network. This guarantees the presence of a filter that hits each input within the effective history [19]. A residual connection was added for each dilated convolution to integrate the convolutional result with the input layer. In this work, the input of the TCN model was defined as *x_t_* and the output was represented by *y_t_*, where *x_t_* contains *n*-dimensional parameters. The output of the TCN model was the *LR* of the next day.

#### 2.3.3. Warning Level Determination

*LR* data were fitted to an isolation forest model [26] to categorize the honey bee daily loss rate according to different warning levels. Isolation forest is an unsupervised learning classification method based on decision trees. It recursively creates partitions by randomly selecting a feature and then picks a random split value between the minimum and maximum value of the selected feature. It produces smaller paths for the outlier values. Unlike other outlier detection methods, isolation forest explicitly identifies anomalies instead of profiling normal data points. Thus, the outlier values can be easily distinguished from non-outlier data. Isolation forest performs better than most anomaly detection algorithms across different datasets, based on receiver operating characteristic (ROC) performance and precision [26] and therefore chosen for the given problematic. Anomalies from *LR* data were detected by isolation forest algorithm and categorized as abnormal, while the rest of the data were classified as normal. Normal level indicates a honey bee population within normal conditions at a natural loss rate without the need of manual interception. An abnormal level suggests detailed monitoring to prevent potential colony damage.

### 2.4. Forecasting Model Training and Optimization Strategy

#### 2.4.1. Data Characterization

Forecasting accuracy depends strongly on the data characteristic. Determining the data characteristic can help in developing an optimized forecasting strategy and selecting a suitable model [27]. Two indices were computed to identify the demand pattern of the data: average inter-demand interval (*ADI*) and square of the coefficient of variation (*CV^2^*). *ADI* is a measure of the demand regularity in time based on the average interval between demands while *CV^2^* is an index that measures the variation in quantities. The two indices were used to classify *LR* data according to four different categories: smooth, intermittent, erratic, and lumpy [28].

#### 2.4.2. Feature Importance Analysis and Selection

The objective of feature selection is to minimize the input features to optimize forecasting performance. *LR* data were fit to a random forest classifier to compute the importance score of each feature using functions provided in Scikit-Learn machine learning library [24]. Afterwards, correlation analysis was carried out to generate a correlation heatmap that describes relationships between features. The correlation coefficient computed between features ranges from -1 to 1, where values higher than 0.5 indicate significant linear correlation. Feature groups (FG) were formed based on the feature analysis results for comparison.

#### 2.4.3. Model Training and Optimization

TCN models were trained based on the different feature groups obtained from the feature selection step for comparison. Grid search was used to find the best values of the hyperparameters, including learning rate, number of epochs, number of TCN layers, batch size, window size, number of filters, kernel size, dilation, and dropout rate. Each model was optimized using the Adam optimizer, while minimizing the mean squared error (*MSE*), with He normal as the kernel initializer [29].

### 2.5. Algorithm Evaluation and Statistical Analysis

To evaluate the performance of the forecasting model, the weighted mean average percentage error (*WMAPE*) was computed. *WMAPE* measures the absolute percentage error in prediction and was calculated by Equation (2): (2)$$WMAPE=\frac{\sum_{i=1}^{N}\frac{\left|y_{i}^{obs}-y_{i}^{pred}\right|}{y_{i}^{obs}}\times 100\times y_{i}^{obs}}{\sum_{i=1}^{N}y_{i}^{obs}}$$
where $y_{i}^{obs}$ denotes the real value, $y_{i}^{pred}$ denotes the predicted value, and *N* is the total amount of data.

There is no standard in determining warnings of the daily population loss rate of honey bee colonies. Thus, the warning level determination method was validated based on the daily population loss rate values found in related literature. In the work of Rumkee et al. [30], it was found that a daily mortality of forager honey bees of more than 15% led to a rapid loss in population and a colony survival rate of only 50%. Meanwhile, Dukas [31] discovered a natural a daily mortality rate of forager honey bees of about 13.4% due to aging and natural deaths. Therefore, here, colonies with *LR* values higher than 13% were considered to be abnormal, for verification purposes. The accuracy of the warning level determination method was computed using Equation (3):(3)$$Accuracy\;\left(\%\right)=100-\frac{\left|A_{predicted}-A_{true}\right|}{A_{true}}\times 100$$
where *A_predicted_* and *A_true_* are the number of predicted abnormal *LR* by isolation forest algorithm and number of true abnormal *LR* greater than 13%, respectively. Student paired *t*-tests (*p* < 0.05) were used to confirm the significant differences between the results and data acquired.

## 3. Results and Discussion

### 3.1. Data Characterization Results

*LR* data obtained from the beehives of each dataset were characterized by computing *ADI* and *CV^2^*, as summarized in Table 3. The results showed a smooth demand pattern for most *LR* data, except for dataset E_B3_. On average, the *ADI* and *CV^2^* of all the datasets indicated a smooth characteristic and the data were regular in time and quantity. High regularity indicates high forecastability [27].

After each experiment, the honey bee colonies were also manually inspected to determine if their condition was normal or collapsed. A colony with thousands of in-and-out activities at the hive entrance was classified to be normal, otherwise, as collapsed. In experiment E_A_, all four beehives were normal while there were two normal beehives (E_B2_ and E_B4_) and two collapsed beehives (E_B1_ and E_B3_) in experiment E_B_. The higher values of *CV*^2^ obtained from datasets E_B1_ and E_B3_ compared to the other data might be indicative. The two collapsed beehives in experiment E_B_ were further analyzed and are discussed in later sections.

### 3.2. Model Input Feature Optimization

#### 3.2.1. Feature Correlation and Importance Analysis

The feature importance analysis results obtained from the data on each beehive are presented in Figure 2. The feature importance scores computed from the data of each beehive varied. In general, it was found that precipitation, hourly precipitation, incoming counts, outgoing counts, and daily count difference had the highest scores. However, it was also found that the collapsed beehives (Figure 2e,g) had different results from the normal beehives (Figure 2a–d,f,h). The feature importance score of normal hives were high in terms of precipitation, precipitation duration, incoming counts, outgoing counts, difference in count (Figure 2a–d,f,h). Meanwhile, the incoming count, difference in count had high scores for collapsed hives (Figure 2e,g), suggesting that the incoming counts and daily count difference were also important for forecasting.

The correlation heatmap of the features from dataset E_A2_ is shown in Figure 3a and is representative for a normal case. The correlation heatmap is highly correlated with the computed feature importance scores (greater than 0.7) for groups of features, such as temperature, humidity, and precipitation, most likely due to their similarities. Highly correlated features are often considered redundant because they do not add useful information to the model. Hence, one feature was selected from each set of related features, forming a feature group called FG2, as shown in Table 4. The computed correlation values of each feature in relation to *LR* were sorted (Figure 3b). It was found that some features had low correlation values of −0.25 to 0.25 such as *WS_gus_*_t_, *WS*, *WD*, *T_min_*, *T_max_*, and *WD_gust_* signifying that these features had neither a positive nor negative correlation to *LR*. Therefore, the features were removed to form another feature group called FG3.

The weather forecast data from the weather station were also utilized to examine their potential effect on the forecasting results. This formed another feature group called FG4, which included the following day’s forecasted values of precipitation, relative humidity, and temperature. Based on the related literature, temperature also has an effect to the behavior of honey bees [32]. Yet, it can be seen from the feature importance score and correlation analysis results that its effect was not considerable for the current datasets. Even so, temperature was still added in all feature groups to ensure that the forecasting model would also be trained to adapt to drastic changes in the environmental condition.

#### 3.2.2. Feature Group Selection

For comparison, a feature group named FG1, which included all the available input features, was prepared. A univariate model, using *LR* data as the only feature, was also trained. Individual forecasting models were trained with a training dataset of 80 days, with the last 20% of the training data as validation set. Other training parameters were not yet considered and were optimized later. The results are shown in Figure 4.

Based on the testing results, the *WMAPE* of the univariate model was about 42.5 ± 3.4%. This extensively high error demonstrates that using only the daily loss rate as input feature is inadequate for forecasting. The multivariate model using FG1 performs better with a *WMAPE* of 20.1 ± 2.8% as supported by *T*-test results with a significant difference between both approaches. There is no significant difference between FG1, FG2, and FG3, this indicates that reducing the number of features does not affect the performance as expected from apparent redundancies. Interestingly, adding the forecasted weather data as an additional set of features improves the performance and reduces the error to 17.1 ± 1.6%. This reflects that forecasted weather-data as supplemental information contribute to an enhanced forecasting model. Based on these findings, FG4 was selected as the input feature group of the forecasting model throughout this work.

### 3.3. Model Training and Hyperparameter Optimization

The dataset was split into training, validation, and testing sets according to time. The training set was used to train the model based on the validation set. The testing set was used for assessing the model performance in forecasting data outside the training and validation set. For normal colonies (E_A1_~E_A4_; E_B2_, E_B4_), the training set (Y_train_) in each time series was the first *N*-day observations, where *N* is the number of training days. The validation set (Y_val_) included the last 20% of the training set. The rest of the dataset was used for testing (Y_test_). For the collapsed colonies (E_B1_, E_B3_), the daily *LR* during the first *N* days were normal and started to become abnormal at day 105 and 90 for E_B1_ and E_B3_, respectively. From prior testing, it was not possible to use normal *LR* data to forecast the abnormal *LR* data. Therefore, a model was trained using the data of E_B1_ for forecasting the data of E_B3_ and vice versa. Similar to the normal colonies, the last 20% of the Y_train_ were used for validation.

The effect of different number of days *N* in training the forecasting model was evaluated to know how much data were needed to attain reliable forecasting performance. TCN models, using FG4 as input feature group, were trained with values of *N*, ranging from 20 to 100, as presented in Table 5. The results show that the model trained with fewer than 80 training days had poor performances with *WMAPE* as high as 50.2%. Using 80 training days and above, it led to better forecasting performance with *WMAPE* as low as 21.6%. Interestingly, it also shows that increasing the value of *N* from 80 to 100 did not considerably improve the forecasting performance. Based on these findings, the number of training days was set to 80 for later analysis.

The results of finding the best model hyperparameters via grid search, using *N* = 80, is shown in Table 6. It was found that the model can perform best by setting the input window size to 10; this means that the model needs at most a length of 10 days to achieve satisfactory forecasting performance. It was also observed that 2 TCN layers were sufficient to yield good forecasting performance; this can be mainly attributed to the smooth characteristic of the data. By applying the tuned hyperparameters, the *WMAPE* of the TCN model was reduced to 16.2%, from 21.6% of the model using the default values.

### 3.4. Daily Population Loss Rate Forecasting Results

By exploiting the optimized model, the long-term forecasting results of the honey bee daily population loss rate for normal, and collapsed beehives are presented in Figure 5 and Figure 6, respectively, while the *WMAPE* boxplots of the forecasting model is shown in Figure 7.

The trained TCN model was able to successfully predict the testing *LR* data of normal beehives (Figure 5). Most importantly, it was able to predict sudden changes in *LR*, as particularly seen in Figure 5a–d. The TCN model performed well on the collapsed beehives by forecasting the sudden changes in *LR* on both datasets, as particularly seen in Figure 6 from day 260 and so on. In general, the TCN model performed well in all datasets with *WMAPE* from 17% to 19% (Figure 7), with paired *t*-tests showing no significant difference between the original and predicted values per normal *LR* dataset. The results also indicate that the selected features in FG4 were sufficient for forecasting, even though there were differences in the condition of the beehives. Paired *t*-test showed a significant difference between normal and collapsed beehives (E_B1_, E_B3_) which most likely was due to the difference in data characteristics as mentioned in Table 3.

### 3.5. Daily Loss Rate Warning Threshold Results

The results of applying isolation forest algorithm on normal beehives to determine abnormal *LR* values are presented in Figure 8 and Figure 9. Figure 8 applies for dataset E_A1_~E_A4_ and Figure 9 applies for datasets E_B1_~E_B4_. It can be easily distinguished that the collapsed beehives (Figure 9e,g) had the most anomalies detected. Particularly, most of the results showed that *LR* values less than 13% were classified as normal otherwise as abnormal. Meanwhile, the results of the normal colonies showed only a few days with abnormal *LR* values. Upon validation, the accuracy of the fitted isolation forest model was about 90.0 ± 8.5%, as shown in Table 7.

## 4. Discussions

The abnormal daily *LR* of the honey bee colonies in this study was about 13%, in agreement with related literature [30,31]. In the work of Rumkee et al. [30], it was found that a colony will collapse if the daily mortality of forager honey bees exceeds 15% and was equally observed for both collapsed beehives (E_B1_, E_B3_) in this study (Figure 9e,g).

Our experiments show that honey bee colony collapses can be predicted. Previous studies identified the collapse of honey bee colonies was most likely caused by in-hive colony behavior and health of the honey bees [31]. A typical in-hive anomaly is caused by the loss of a queen bee. Indicators of a queen-less colony include missing eggs and brood, increased number of drones, a significant drop in population, and stored pollen and honey in the brood cells. These indicators were observed from the two collapsed beehives in the present study. Based on our recorded data, the loss of the queen occurred at about day 105 of E_B1_ and day 95 of dataset E_B3_. According to Lopes et al. [33], workers of queen-less colonies can live up to 80 days. This was similarly observed in our study which shows that the colony collapsed about 60–80 days after the queen disappeared (Figure 6a,b).

There are no strict conventions for defining the *LR* threshold of a collapsed honey bee colony. Based on the experimental results obtained, the daily loss rate of the beehives became abnormal or several consecutive days, which was about 50 and 60 days for dataset E_B1_ and E_B3_, respectively. The presence of an abnormal status as automatically classified is a strong indication for the required interference of beekeepers at early stage to prevent further damage to the honey bee colony.

## 5. Conclusions

A reliable method for forecasting and early detection of abnormal honey bee colony population loss rate is presented. The proposed method was optimized by appropriately selecting informative features and tuning the hyperparameters of a TCN forecasting model. The forecasting model performed best by using selected features, such as daily population loss rate, incoming counts, difference in counts, temperature, humidity, precipitation, and the forecasted following day’s temperature, humidity, and precipitation obtained from the local weather station. Upon comparison with other selected feature groups, it was discovered that using the forecasted weather data as supplemental input feature improved the model performance. It was also found that the forecasting model performed well by training it with at least 80 days of historical population loss rate data. The forecasting model was able to accurately forecast the following day’s honey bee colony population loss rate with a *WMAPE* of 17% to 19%. The forecasting model optimization strategy can be used as a reference for researchers to effectively improve their model.

The population loss rate data output of the forecasting model was further utilized to define the warning levels using an isolation forest algorithm, thus enabling the proposed method to determine whether the beehive colony status to be normal or abnormal. By validating the obtained values with the results of related works, it was found that the warning level determination method was able to yield an accuracy of 90.0 ± 8.6%.

Although the exact reason why a honey bee colony collapsed remained unclear, the data of the image monitoring system were successfully utilized to train a forecasting model that can potentially explain the phenomenon. If the daily loss rates were abnormal for a certain number of consecutive days, it indicated that honey bee collapse may potentially occur. The proposed method, together with the image monitoring system, was proven to assist in the early detection of collapsed honey bee colonies. This work can be used to deliver fast and reliable data-driven information to honey bee colony managers so that they can ensure the health of their honey bee colonies.

To actualize the system, an early warning can be determined by detecting abnormal daily loss rates while a notification can be sent to the beekeeper via mobile phone. Upon receiving the message, beekeepers can check the beehive and prevent potential harm to the honey bee colony. Beekeepers can install the monitoring system in selected beehives and monitor the beehive health index remotely.

## Figures and Tables

**Figure 1 sensors-21-03900-f001:**
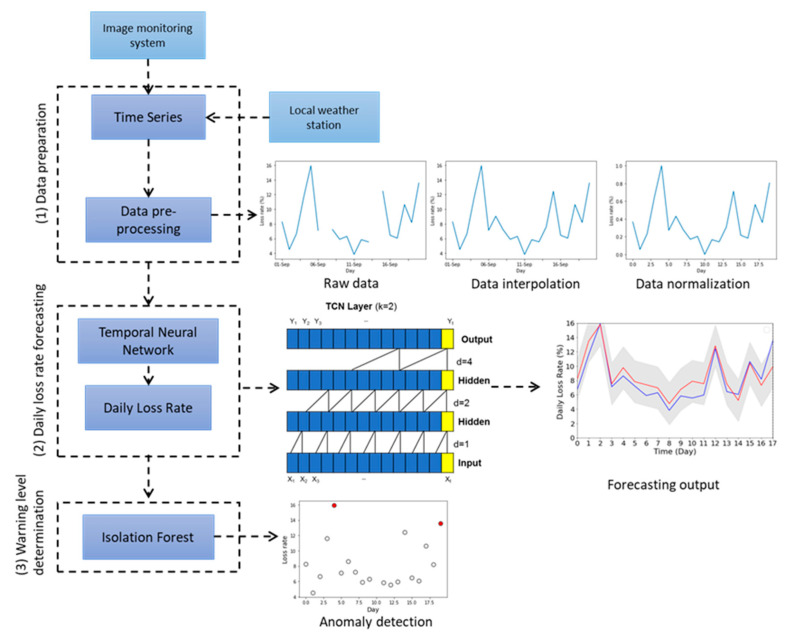
Flowchart of the honey bee colony daily population loss rate forecasting and early warning algorithm.

**Figure 2 sensors-21-03900-f002:**
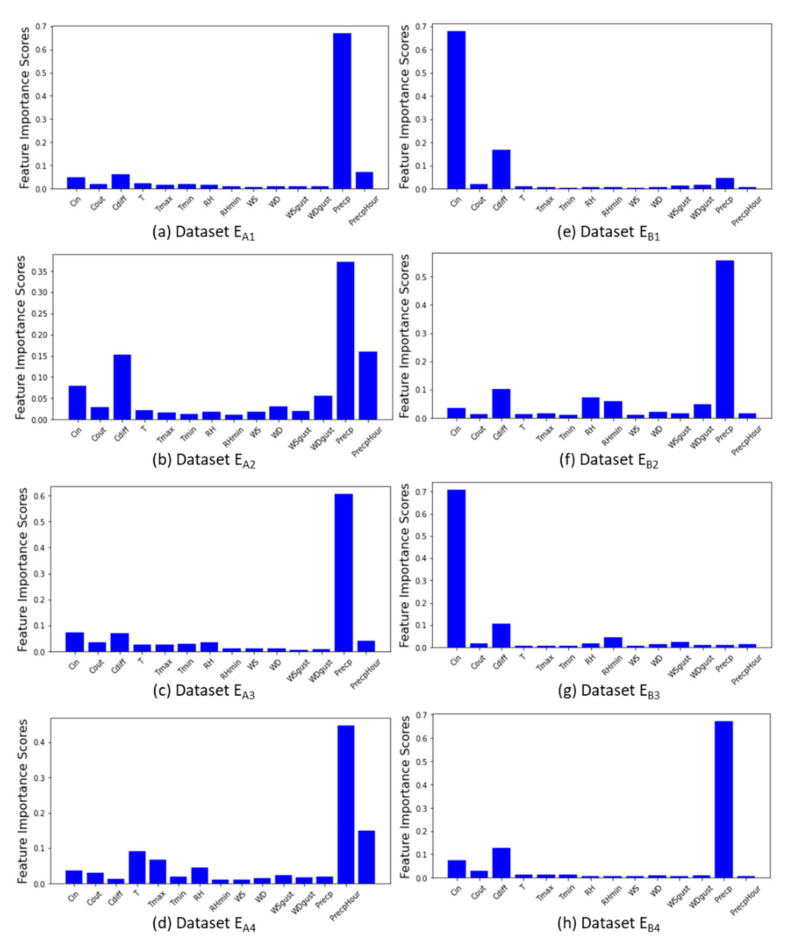
Feature importance score analysis results obtained from each dataset.

**Figure 3 sensors-21-03900-f003:**
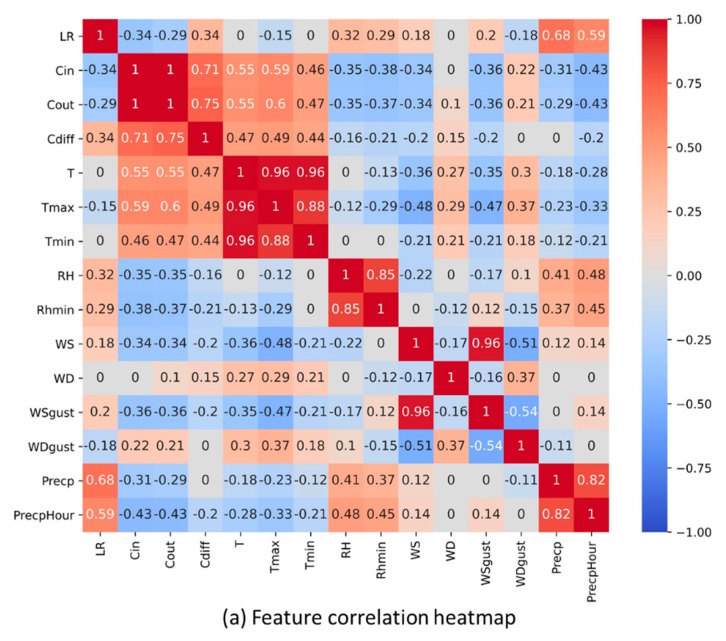
Correlation heatmap and sorted *LR* correlation values of dataset E_A2_.

**Figure 4 sensors-21-03900-f004:**
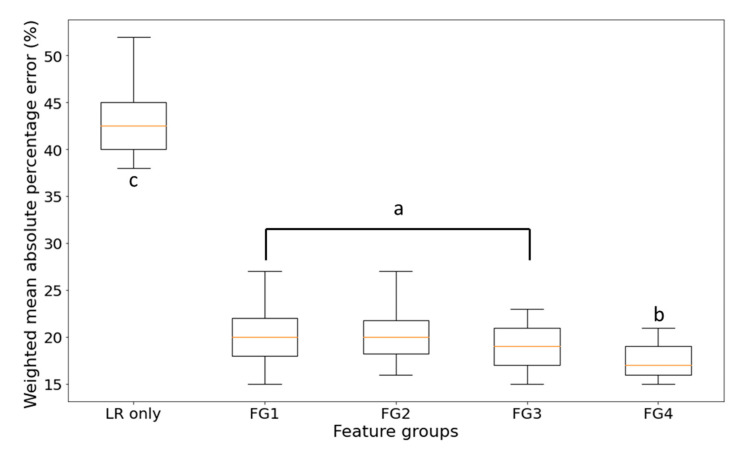
TCN forecasting model *WMAPE* boxplots using different input feature groups. Letter a, b and c denotes the significant different between groups.

**Figure 5 sensors-21-03900-f005:**
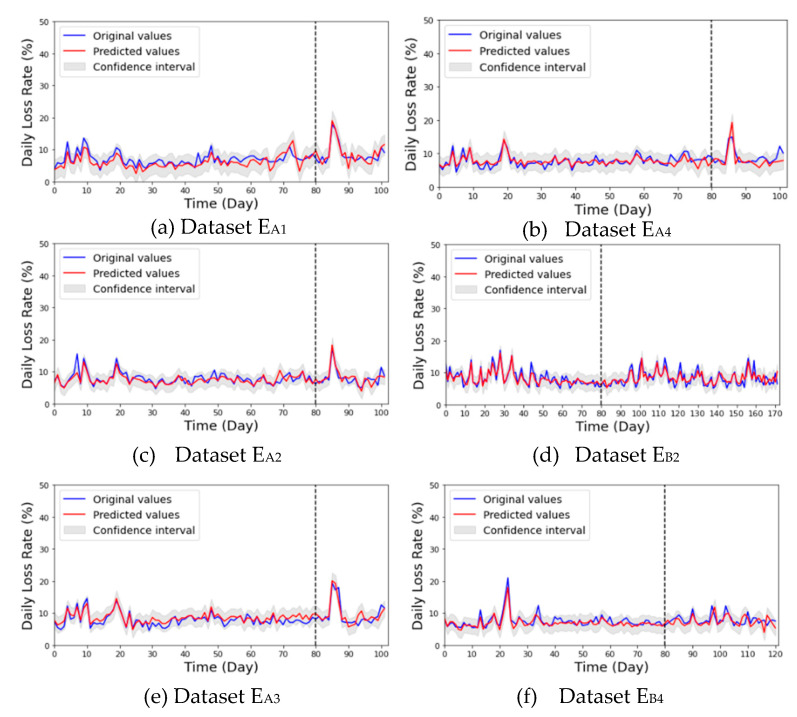
Daily population loss rate forecasting results of normal beehives (E_A1_~E_A4_; E_B2_, E_B4_). The data are separated using dashed lines indicate the training (**left**) and testing (**right**) data.

**Figure 6 sensors-21-03900-f006:**
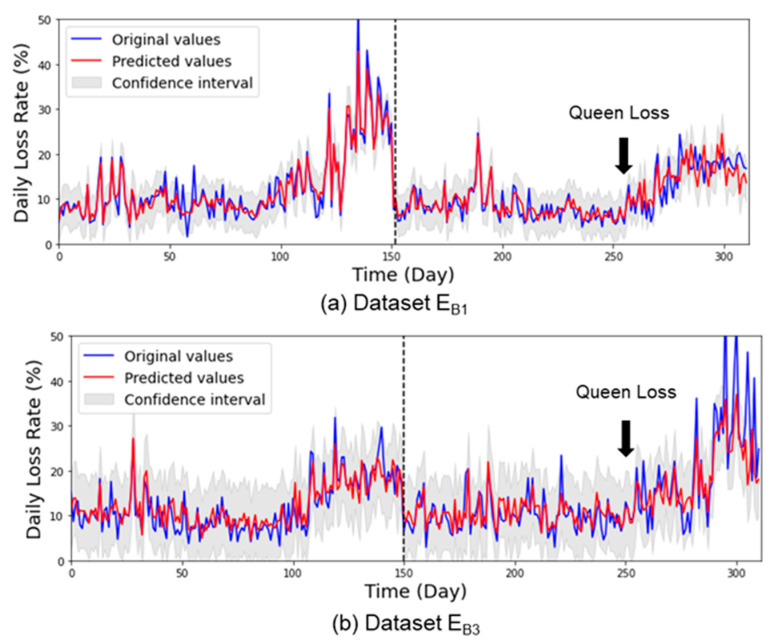
Daily population loss rate forecasting results of collapse beehives (E_B1_, E_B3_). The data are separated using dashed lines indicate the training (**left**) and testing (**right**) data.

**Figure 7 sensors-21-03900-f007:**
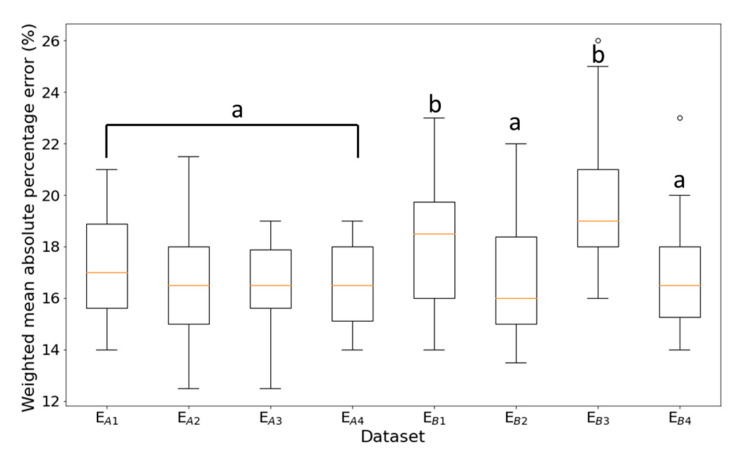
TCN forecasting model *WMAPE* boxplot for each *LR* dataset, where letters a and b indicate the significantly different groups.

**Figure 8 sensors-21-03900-f008:**
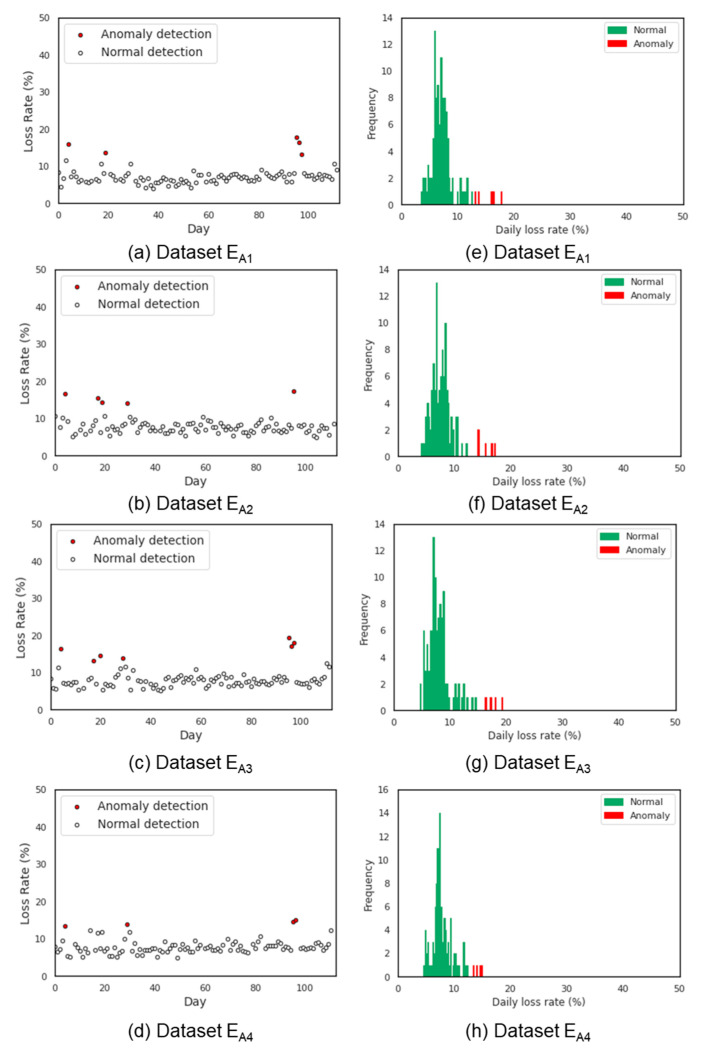
*LR* anomaly detection results (**a**–**d**) and relative frequency of the *LR* data (**e**–**h**) obtained from dataset E_A_.

**Figure 9 sensors-21-03900-f009:**
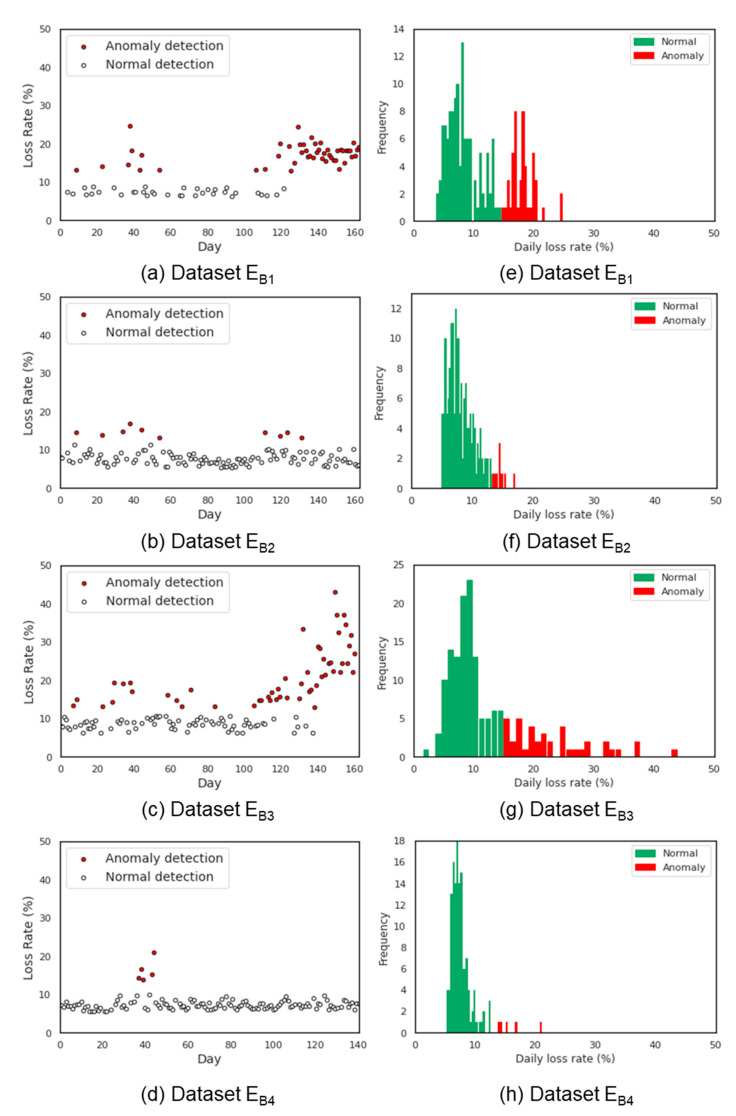
*LR* anomaly detection results (**a**–**d**) and relative frequency of the *LR* data (**e**–**h**) obtained from dataset E_B_.

**Table 1 sensors-21-03900-t001:** Forecasting model input data features collected from the monitoring system and the local weather station.

Feature Name	Description	Units
*LR*	daily loss rate	%
*c_in_*	daily incoming count	-
*c_out_*	daily outgoing count	-
*c_diff_*	daily difference	-
*t*	ambient temperature	Celsius
*t_max_*	maximum temperature	Celsius
*t_min_*	minimum temperature	Celsius
*rh*	relative humidity	%
*rh_min_*	minimum humidity	%
*ws*	wind speed	m/s
*wd*	wind direction	degree
*ws_gust_*	wind speed gust	m/s
*wd_gust_*	wind direction gust	degree
*precp*	precipitation	mm
*precphour*	duration of precipitation	hour

**Table 2 sensors-21-03900-t002:** Information on each experiment.

Experiment#	Start Date	End Date	# of Hives	Duration
E_A_	2019/08/08	2019/12/08	4	122 days
E_B_	2020/04/14	2020/09/27	4	169 days

**Table 3 sensors-21-03900-t003:** Summary of data characterization results.

Dataset	Duration	*ADI*	*CV^2^*	Characteristic	Condition
E_A1_	112 days	1.03	0.33	Smooth	Normal
E_A2_	112 days	1.04	0.29	Smooth	Normal
E_A3_	112 days	1.03	0.32	Smooth	Normal
E_A4_	112 days	1.05	0.25	Smooth	Normal
E_B1_	171 days	1.06	0.47	Smooth	Collapsed
E_B2_	182 days	1.07	0.30	Smooth	Normal
E_B3_	161 days	1.06	0.64	Erratic	Collapsed
E_B4_	141 days	1.05	0.28	Smooth	Normal

**Table 4 sensors-21-03900-t004:** Information on each feature group.

Group Name	List of Features
FG1	*LR, C_in_, C_out_, C_diff_, T, T_max_, T_min_, RH, RH_min_, WS, WD, WS_gust_, WD_gust_, Precp, Precp_hour_*
FG2	*LR, C_in_, C_diff_, T, RH, WS, WD, WD_gust_, Precp*
FG3	*LR, C_in_, C_diff_, T, RH, Precp*
FG4	*LR, C_in_, C_diff_, T, RH, Precp, T_t+1,_ RH_t+1_, Precp_t+1_*

**Table 5 sensors-21-03900-t005:** Forecasting performance using different number of training days.

	Number of Training Days *N*				
	20	40	60	80	100
*WMAPE* (%)	50.2	45.1	27.4	21.6	21.1

**Table 6 sensors-21-03900-t006:** Optimized model hyperparameters found by grid search.

Hyperparameter Name	Range of Value/s	Optimized Value/s
Learning rate	[0.1, 0.01, 0.001, 0.0001]	0.001
Number of epochs	[30, 50, 70, 90, 110]	70
Optimizer	[Adam]	Adam
Loss function	[*MSE*]	*MSE*
Number of TCN layers	[1, 2, 3]	2
Batch size	[1, 5, 10, 15]	10
Window size	[1, 5, 10, 15]	10
Number of filters	[8, 16, 32, 64]	32
Kernel size	[1, 2, 3, 4]	2
Dilation	[[1, 2, 4], [1, 2, 4, 8]]	[1, 2, 4]
Activation	[ReLu]	ReLu
Dropout rate	[0.2, 0.3, 0.4, 0.5]	0.2
Kernel initializer	[He normal]	He normal

**Table 7 sensors-21-03900-t007:** Isolation forest algorithm performance on detecting abnormal *LR* values.

Dataset	Real	Predicted	Accuracy (%)
E_A1_	5	4	80.0
E_A2_	5	4	80.0
E_A3_	7	7	100
E_A4_	4	4	100
E_B1_	57	52	91.2
E_B2_	10	8	80.0
E_B3_	53	48	90.6
E_B4_	5	4	80.0

## Data Availability

The data that support the findings of this study are available from the corresponding author, T.-T.L., upon reasonable request.

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
