# Peer review of "Honey Bee Colony Population Daily Loss Rate Forecasting and an Early Warning Method Using Temporal Convolutional Networks"

_sensors, 2021, doi:10.3390/s21113900_

Round 1

Reviewer 1 Report

The goal of this study is very interesting.

The study provides researchers with an excellent tool for being able to discern when a colony begins to fail and the ability to get closer to the time when the event(s) precipitating collapse occur. This increases the possibility that the actual cause of the colony failure will be easier to identify.

It is possible that some beekeepers may be able to use the monitoring method described in this manuscript to more closely watch the health of their colonies.  Unfortunately, it is likely this monitoring method is too expensive for many beekeepers. 

The information provided in the Methods and Materials section was very organized.

As an aside, the proper noun-verb agreement when the word data was used was much appreciated.

Reviewer 2 Report

Honey Bee Colony Population Daily Loss Rate Forecasting and Early Warning Method Using Temporal Convolutional Networks

The authors present an automatic method to estimate honey bee hive health by predicting the population daily loss rate using temporal convolutional neural networks.

The method presented relies heavily in previously published work for bee tracking and activity monitoring. Nevertheless, the paper presents enough novelty to stand on its own. The work does not present major novelties in terms of deep learning but it does present an interesting example on how to use this technology to solve applied problems. Furthermore, building on previous work results in the research presenting a complex solution to a very interesting practical problem. The paper presents several experiments that are convincing and well developed, but there are some methodological and language issues that need to be addressed before the paper is accepted for publication.

MAJOR COMMENTS

I am slightly baffled at the duality of the analysis of the importance of features by using random forests and the use of TCN to actually predict using those features. I wonder if it would not have been more effective to just use random forests for prediction too. Alternatively, wouldn't it be possible to include the features considered in the "parameter space grid search" summarized in table 6?

The experiments are interesting but I believe they are written in a slightly confusing way. Please make sure to state what you hope to show with each of them, how you intend to evaluate your results and to cite specific numbers that exemplify any point that you make. Furthermore, some important information is missing. The most important part is that the division between testing, training and validation data is unclear. FIgure 5 mentions training and testing sets but I have not been able to find what these stand for. Should the testing and training sets be built by randomly choosing from the whole of the data, this may be a major methodological problem.

Please make sure to specify how you built your training and testing sets. The following principles should be followed always:

1) The testing and training sets should be clearly separate. In this case, it would make sense to use data from some of the colonies to train and some of them to test. Usually leave-one-out cross correlation approaches are used in these cases to maximize data and avoid problems. The testing set should not be part in any way of the training process (and this includes choosing the best network hyperparameters). If I understand correctly, you used some time steps to train and others to test, I believe this may be producing results that look better than their real performance in practice.

2) Consider using a validation set, separate from the testing to get some indication of possible overfitting problems. The very high number of epochs used to train the model as well as the behavior seen in figure 5 suggests that you have overfitting problems.

MINOR COMMENTS

Introduction

The first paragraph of the introduction is a bit difficult to read because of wordy language and some grammar problems, please re-write.

The language of the paper presents some mild problems that, at times, make it slightly difficult to follow the precise details of what is being said. I will  provide a few examples but the authors should work hard at improving the text as in its current state it detracts from the work done. Perhaps the most frequent problem is the use of unnecessary and "wordy" language, please be concise and make your point succinctly. 

Material and Methods:

"Convolutions are causal; thus, the model does not depend on any future timestep data."

Results

Figure 4, "Groups with statistical significance of 5%" does not have a precise meaning, please be rigourous.

The number of epochs used seems extremely long for the type of data that you are using, 

Conclusions:

Please expand on how your method can help understand hive collapse and how exactly you envision beekepers using it.

Language problems:

36,(grammar) To maintain honey bees are healthy and thriving,

43, In the recent decades, ("the" is incorrect)

50,51, unnecessarily complex sentence (wordy) : "that can provide beekeepers with more valuable information and assist beekeepers in beehive management"

As much as possible, please refrain from using "they" to mention other authors, it is often informal.

97,98, grammar: "Unfortunately, there are almost no defining warning levels for honey bee activity,which led to the presented method in this paper."

180, grammar "The TCN forecasting model consisted of a series of blocks in which each block contained a sequence of convolutional layers"

Round 2

Reviewer 2 Report

My previous concerns have been sufficiently addressed.